# Colchicine pre-treatment and post-treatment does not worsen bleeding or functional outcome after collagenase-induced intracerebral hemorrhage

**Cassandra M. Wilkinson**[1©], **Aristeidis H. Katsanos**[2©], **Noam H. Sander**[3©], **Tiffany F. C. Kung**[1], **Frederick Colbourne**[1,3‡]*, **Ashkan Shoamanesh**[2‡]

1 Department of Psychology, University of Alberta, Edmonton, Alberta, Canada, 2 Department of Medicine (Neurology), McMaster University/Population Health Research Institute, Hamilton, Ontario, Canada, 3 Neuroscience and Mental Health Institute, University of Alberta, Edmonton, Alberta, Canada

© These authors contributed equally to this work.
‡ FC and AS also contributed equally to this work.
* fcolbour@ualberta.ca

**Data Availability Statement:** All relevant data are within the paper and its Supporting Information files.

## Abstract

Patients with intracerebral hemorrhage (ICH) are at increased risk for major ischemic cardiovascular and cerebrovascular events. However, the use of preventative antithrombotic therapy can increase the risk of ICH recurrence and worsen ICH-related outcomes. Colchicine, an anti-inflammatory agent, has the potential to mitigate inflammation-related atherothrombosis and reduce the risk of ischemic vascular events. Here we investigated the safety and efficacy of colchicine when used both before and acutely after ICH. We predicted that daily colchicine administration would not impact our safety measures but would reduce brain injury and improve functional outcomes associated with inflammation reduction. To test this, 0.05 mg/kg colchicine was given orally once daily to rats either before or after they were given a collagenase-induced striatal ICH. We assessed neurological impairments, intra-parenchymal bleeding, Perls positive cells, and brain injury to gauge the therapeutic impact of colchicine on brain injury. Colchicine did not significantly affect bleeding (average = 40.7 μL) at 48 hrs, lesion volume (average = 24.5 mm³) at 14 days, or functional outcome (median neurological deficit scale score at 2 days post-ICH = 4, i.e., modest deficits) from 1–14 days after ICH. Colchicine reduced the volume of Perls positive cells in the perihematomal zone, indicating a reduction in inflammation. Safety measures (body weight, food consumption, water consumption, hydration, body temperature, activity, and pain) were not affected by colchicine. Although colchicine did not confer neuroprotection or functional benefit, it was able to reduce perihematomal inflammation after ICH without increasing bleeding. Thus, our findings suggest that colchicine treatment is safe, unlikely to worsen bleeding, and is unlikely but may reduce secondary injury after an ICH if initiated early post ICH to reduce the risk of ischemic vascular events. These results are informative for the ongoing CoVasc-ICH phase II randomized trial (NCT05159219).

**Funding:** CMW is supported by a Canadian Institutes of Health Research scholarship and a Killam trust scholarship. NHS is supported by a Branch Out Neurological Foundation scholarship. TFCK is supported by a Canadian Institutes of Health Research scholarship. AHK is supported by a career award from the Department of Medicine, McMaster University. AS is supported by the National Institutes of Health, Heart and Stroke Foundation of Canada, the Marta and Owen Boris Foundation and the Population Health Research Institute. The funders had no role in study design, data analysis, or decision to publish.

**Competing interests:** The authors have declared that no competing interests exist.

## Introduction

Survivors of intracerebral hemorrhage (ICH) have an increased risk of recurrent ICH and future ischemic vascular events [1–3]. Antithrombotic therapy is the mainstay of treatment to prevent ischemic vascular events in patients with stroke and heart disease. However, antithrombotic agents increase the risk and severity of ICH [4, 5]. Current guidelines [6] recommend cessation of antithrombotic medication in patients with acute ICH and there exists substantial uncertainty about if or when they should be resumed in ICH survivors. This likely contributes to an increased risk of ischemic vascular events in ICH patients, particularly in the first months after the ictus [7] when physicians are most reluctant to re-introduce them [8]. Thus, there is an unmet need to identify medications that can prevent thrombotic events without increasing the risk of bleeding in ICH survivors.

Inflammation contributes to plaque formation and increased plaque instability [9]. Colchicine, a U.S. Food and Drug administration approved microtubule inhibitor, reduces inflammation, and specifically prevents leukocyte motility, which relies on microtubule actions [9, 10]. Reducing inflammation is a promising treatment target with which to reduce ICH damage. Accordingly, there have been multiple trials testing chronic colchicine treatment to reduce the rate of major adverse cardiovascular events, including ischemic stroke, in patients with coronary artery disease [10–12]. These trials showed the rate of cardiovascular events, including ischemic stroke, was reduced in patients given 0.5 mg colchicine once daily. Thus, colchicine could provide a novel treatment option for ameliorating the thrombotic risk in ICH survivors, particularly in the acute to sub-acute phase after ICH, when antithrombotic therapy is least likely to be prescribed.

Colchicine can potentially reduce inflammation, a contributor to secondary injury after ICH, to lessen injury and promote recovery. In response to extravasated blood, microglia quickly respond to initiate an inflammatory cascade [13, 14] that persists for days to weeks [15]. Inflammation helps by clearing blood and cellular debris; however, inflammation also increases oxidative stress, blood-brain barrier damage, and cell death [13, 16]. If colchicine reduces this inflammation acutely after ICH, this may lead to neuroprotection and improved outcomes while preventing future ischemic vascular events. However, colchicine is a known modulator of platelet function and has been suggested to impair platelet aggregation, decreasing hemostasis [17]. Therefore, colchicine given early after ICH could potentially increase bleeding and worsen outcomes. It is important to consider early intervention after stroke, as the Colchicine Cardiovascular Outcomes Trial (COLCOT) found time-to-treatment followed a U-dose response curve. In this trial, earlier (0–3 days) and later (24–30 days) post-myocardial infarction treatment was associated with a greater risk reduction, indicating the importance of treatment timing [18]. Moreover, treatments targeting secondary inflammatory brain injury need to be initiated early post-ICH to maximize their likelihood of success.

We assessed the safety and efficacy of colchicine in rat during the acute phase of collagenase-induced ICH [19, 20], a common model [21]. We chose this model as bleeding progresses over hours, and thus, any coagulopathy induced by colchicine is more likely to be detected. This represents the worst-case situation of a patient that experiences hematoma expansion [22]. Here, we looked at two dosing timings: a pre-ICH treatment group, representing a clinical population that is already taking colchicine at the time of their ICH, and a post-ICH treatment group, representing patients being prescribed colchicine acutely after ICH, both to reduce inflammation and to prevent future ischemic vascular events. In experiment one, we tested the safety of our colchicine dose in naïve animals by assessing the effects of a two-week dosing paradigm on temperature, activity, pain, feeding behaviours, and weight to see if colchicine caused fever, lethargy, pain, or gastrointestinal problems. We hypothesized that our

dosage would not cause any safety concerns. For experiment two, we assessed the effect of colchicine on neurological deficits, injury volume, and inflammation at 48 hours post-ICH. We hypothesized that colchicine would reduce inflammation along with attenuating brain injury volume and neurological impairment. Third, we hypothesized that colchicine would lessen edema and neurological impairments at 48 hours post-ICH, which is approximately when edema peaks in our model [23–25]. Last, we measured neurological deficits, walking ability, and lesion volume at 14 days post-ICH to determine whether colchicine provides lasting benefits, as hypothesized.

## Methods

This study is reported in accordance with the Animal Research: Reporting of *In Vivo* Experiments (ARRIVE) guidelines for animal research [26]. All experimental procedures were performed at the University of Alberta according to the Canadian Council on Animal Care Guidelines, and were approved by a local Animal Care and Use Committee (protocol AUP960).

Experiments 1, 2, and 4 were planned *a priori* to improve translational rigor. Experiment 3 was not pre-planned, but as inflammation can worsen edema it was important to assess edema at 48 hours post-ICH as a follow-up to experiment 2 [14].

### Subjects

Eighty-four male Sprague Dawley rats were obtained from Charles River Laboratories (Saint Constant, Quebec). Each single rat was considered an experimental unit. Animals weighing between 350 and 450 g (~3–4 months old) were used for our first and third experiments; our second and fourth experiments used animals weighing between 250 and 350 g (~2–3 months old). The slight difference in weight between experiments was simply to accommodate experimenter schedules. All animals were singly housed during experiments to ensure rats only consumed their own dose and thus avoid confounds due to the drug administration method. Food (Purina rodent chow) and distilled water were provided *ad libitum*. Animals were kept in temperature and humidity-controlled rooms with 12 hr light cycles (12 hrs light, 12 hrs dark). All experimental procedures were done during the light phase.

### Experimental design

In all experiments, except for the first as it had only one group, rats were assigned to groups using a random number generator (random.org). Treatments were prepared by another researcher to ensure blinding. All rats were handled for 10 minutes on two separate occasions prior to baseline behavioural testing to gain familiarity with experimenters. Food and water consumption and body weight were measured daily in all experiments to continually monitor animal health.

**Experiment 1.** This experiment was designed as a pilot study to give insight into the safety of our colchicine dose in our animal model. Four rats were given a daily oral dose of 0.05 mg/kg colchicine for 14 days. Temperature and activity were monitored continually starting at baseline (prior to colchicine treatment) using core telemetry probes. Food and water consumption and body weight were monitored daily, and pain was assessed using the rat grimace scale (RGS). This experiment did not include a control group as the expected parameters are well known in healthy animals, and baseline measures were obtained for reference. A change in temperature of more than 0.5°C, a decrease in food or water consumption of more than 20%, or an increase of more than 2 units on the RGS were set as thresholds for concern. Pain

assessments were blinded to time, but other assessments were not blinded as rats were all subjected to the same treatment.

**Experiment 2.** Hematoma volume was our primary endpoint for this experiment, which we assessed histologically at 48 hours after collagenase infusion. Thirty rats were randomized into three treatment groups (n = 10/group): control vehicle treatment (VEH), colchicine treatment starting 2 days prior to ICH (COLC-PRE), or colchicine treatment starting 2 hours after ICH (COLC-POST). Behavioural endpoints were assessed at baseline (3 days prior to ICH) and at euthanasia (48 hours post-ICH); hematoma volume and neurons in the peri-hematoma zone (PHZ) were also quantified histologically.

**Experiment 3.** Brain water content, a measure for edema, was our primary endpoint for this experiment. Twenty rats (n = 10/group) were randomized into either the VEH or COLC-POST (treatment initiated 2 hours after ICH) groups. Brain water content was assessed at 48 hours post-ICH. Behavioural endpoints were assessed at baseline (2 days prior to ICH) and euthanasia (48 hours post-ICH).

**Experiment 4.** Neurological deficit scale (NDS) and horizontal walking test scores were combined into a composite behavioural score as our primary endpoint for this experiment. Thirty rats (n = 15/group) were randomized into either the VEH or COLC-POST groups. Behavioural assessments were performed at baseline (1 day prior to ICH) and at 3, 7, and 14-days post-ICH. Our secondary endpoint (lesion volume) was measured at 14 days post-ICH to quantify stroke severity.

## Colchicine dosage

Every 24 hours from treatment initiation to euthanasia, 0.05 mg/kg colchicine (Sigma Aldrich, product number C9754) in saline was provided for voluntary oral consumption. To ensure clinical relevancy, this dose was determined to be comparable to the common dose used in patients with coronary artery disease (0.5 mg daily) based on allometric scaling [10, 27], and is within the range used in other preclinical studies aimed at reducing inflammation (0.02–0.75 mg/kg) [28–30]. Although our dose was on the lower end of preclinical doses used, we chose this dose based on a clinically feasible dose, which is kept low due to safety concerns [31]. Doses were mixed into Pillsbury™ sugar cookie dough for rats to consume. As rats are neophobic, plain cookie dough without colchicine was given prior to the start of dosing to expose them to our method of delivery. In the COLC-POST treatment group, colchicine cookie dough was delayed to 2 hours post-ICH (experiments 2–4) to match the clinical feasibility of colchicine administration after ICH. In experiment 2, the COLC-PRE group started treatment 2 days before ICH surgery to model the clinical patient population that is already prescribed colchicine before stroke onset.

## Telemetry probe implantation

In the first experiment, four rats were anesthetized with isoflurane (4% induction, ~2% maintenance in 60% $N_2O$, remainder $O_2$) and had sterile telemetry probes (Model TA10TA-F40, Data Sciences International, St. Paul, MN) implanted into the peritoneal cavity to monitor temperature and activity as done previously [32]. These probes were calibrated to ± 0.2˚C accuracy by comparison to a digital thermocouple probe and calibration grade mercury thermometer. Abdominal incisions were closed using Vicryl sutures (Ethicon, Somerville, NJ, USA). Bupivacaine hydrochloride (0.5 mg/kg, Steri*Max* Inc., Oakville, ON) and meloxicam (0.2 mg/kg, Boehringer Ingelheim (Canada) Ltd., Burlington, ON) were injected once subcutaneously (S.C.) for analgesia at the time of surgery. All probe implantations were performed 4 days before the start of colchicine treatment to reduce surgical confounds.

## Intracerebral hemorrhage

In experiments 2–4, ICH was induced using the well-established collagenase model as previously described [19, 20, 32]. Rats were anesthetized with isoflurane (4% induction, ~2% maintenance in 60% $N_2O$, remainder $O_2$), and a hole was drilled in the skull (0.5 mm anterior, 3.5 mm lateral to Bregma) [33]. To produce a moderately sized ICH, 1.0 μL of bacterial collagenase solution was infused into the left striatum (Type IV-S, Sigma, 0.6 U/μL; 6.5 mm below skull surface). A small metal screw was used to seal the hole and the incision was stapled closed. Prior to surgery and after collagenase infusion, bupivacaine hydrochloride (0.5 mg/kg S.C.) was administered for pain relief. During surgery, rectal temperature was maintained around 37˚C using a water-heated pad placed under the animal.

## Behavioural assessments

Rats were left in the behavioural testing room for 30 minutes to acclimatize to the new setting before all behavioural training and assessments.

**Grimace scale.** The rat grimace scale was used to assess any pain due to colchicine treatment in our first experiment [32, 34]. Rats were videotaped for 5–10 minutes before colchicine treatment (baseline), 1 day, and 14 days after colchicine treatment was initiated. Ten screen capture images were taken from each video at regular intervals. For each image, orbital tightening, nose/cheek flattening, ear changes, and whisker changes were scored separately from 0 to 2 and then later combined to give a score from 0 to 8, with 0 indicating no pain.

**Neurological Deficit Scale (NDS).** All rats in experiments 2–4 were trained and tested on the NDS tasks including: spontaneous circling in a clear cylinder, bilateral forepaw grasping ability, beam walking, contralateral hindlimb retraction, and bilateral forepaw flexion. Each were scored on a scale from 0–3, except forepaw flexion that was scored from 0–2. These tasks were performed as done previously, with all categories compiled to give a total score between 0 (no deficits) to 14 (most severe deficits) [35]. Animals were trained on each component once before baseline testing.

**Horizontal ladder walking test.** Animals in the fourth experiment were trained and tested on the horizontal ladder walking test [35]. Rats were videotaped while walking across a 0.5 m long segment of a 1 m long ladder composed of variably spaced horizontal rungs (1–3 cm apart) [35]. After two training sessions, baseline assessments were made 1 day before ICH and further assessments were performed at 3, 7, and 14 days after ICH. To assess walking ability, we counted the total number of times that each paw or limb missed the rungs across two trials on each testing day.

## Histology

Animals in experiments 1, 2, and 4 were euthanized using a lethal dose of sodium pentobarbital (100 mg/kg I.P.) and transcardially perfused with 0.9% saline, followed by 10% neutral buffered formalin. Brains were collected and post-fixed in formalin prior to Vibratome (Leica VT1200 S) sectioning. Coronal sections (80 μm thick) were taken every 360 μm and stained using cresyl violet; the area of sectioning encompassed the entire hematoma or lesion. In experiment 2, hematoma volume was calculated manually using ImageJ 1.53a (National Institutes of Health, USA). The calculation can be simplified as: hematoma volume = average area of hematoma × interval between sections × number of sections. The number of neurons in the dorsal PHZ under the corpus callosum were counted in representative images from each section using Image J. In experiment 4, total tissue loss was calculated using ImageJ as done previously [35]. This method calculates total tissue loss by subtracting the volume of the remaining tissue in the injured hemisphere from the volume of the intact hemisphere, after accounting

for cavity formation and atrophy. The volume of each hemisphere was calculated as: hemisphere volume = average (total hemisphere area–area of ventricle–area of damage) × interval between sections × number of sections.

In experiment 2 and 4, an additional set of sections were subjected to Perls Prussian Blue staining to detect ferric iron [36–38]. Sections were incubated in a solution composed of equal volumes of 4% potassium ferrocyanide and 4% HCl for 15 minutes, then washed in distilled water for 4 minutes before being counter-stained with 0.125% neutral red. Three images from the PHZ zone (dorsal, ventral, and lateral) were taken from the section of maximal hematoma volume. Using ImageJ, the number of iron-positive macrophages in each image were counted and averaged as a measure of inflammation. In experiment 2, there were too few Perls positive cells to quantify, likely due to the early timing post-ICH and limited ferric iron within inflammatory cells. In experiment 4, due to the variable distribution of Perls positive cells across animals (e.g., narrow bands of cells vs. evenly distributed cells), it was difficult to image a consistent region of interest for all animals. Thus, as an additional measure of inflammation, the volume of Perls positive tissue was assessed using the formula: area of Perls positive tissue × interval between sections × number of sections.

## Muscle water content

During euthanasia, abdominal muscle tissue samples were collected from all animals in experiment 2 prior to formalin fixation. Tissue was weighed prior to baking in the oven at 100°C for 24 hours. The samples were weighed again after baking and muscle water content was calculated as (wet weight–dry weight) / wet weight × 100% and used as a measure of hydration [39].

## Brain water content

Animals in experiment 3 were anesthetized with isoflurane (4% in 60% $N_2O$, remainder $O_2$) 48 hours after ICH and quickly decapitated. Brains were harvested and blocked from 2 mm anterior to 4 mm posterior of the site of collagenase infusion to isolate striatum and cortex from both hemispheres. Tissue was weighed immediately before baking at 100°C for 24 hours, after which the tissue was weighed again. Brain water content was calculated as (wet weight–dry weight) / wet weight × 100% [39]. Above-normal brain water content arises from serum extrusion from the clot as well as vasogenic edema.

## Statistical analysis

Sample sizes were determined *a priori* based on previous work and statistical power calculations. For experiment 2, we determined 10 rats/group had 80% power to detect a 30% difference in hematoma volume based on an expected standard deviation [SD] = 4.5 μL based on previous experimental data [40]. Group sizes of 10 rats/group were calculated to give 80% power to detect a mean difference of 1% in brain water content (SD = 0.75) for experiment 3 [40]. We calculated group sizes of 15 rats in experiment 4 would give 80% power to detect a 35% difference in composite behavioural score (SD = 7.75) based on previous experimental data [35].

GraphPad Prism (v 6.0, GraphPad Software Inc, La Jolla, CA) statistical software was used to analyze all data. All parametric data is presented as mean ± 95% confidence interval (C.I.), while ordinal data (NDS scores) are presented as median ± interquartile range (IQR). To ensure test assumptions were satisfied, the Brown-Forsythe test and the F-test were used to compare variances for one-way analysis of variance (ANOVA) and two-way ANOVA, respectively. The Shapiro-Wilk test was used to test for normality. Temperature, activity, and food consumption data from experiment 1 were analyzed using a one-way repeated measures

ANOVA, and a two-way ANOVA was used to analyze body weight data from experiment 2. A mixed-effects analysis was required to assess body weight (experiment 1), food consumption (experiment 2), and water consumption (experiments 1 and 2) due to missing data. Muscle water content (experiment 2) was analyzed using a one-way ANOVA while the Kruskal-Wallis test was used for ordinal NDS data in experiment 2. Tukey's test and Sidak's correction were planned as post-hoc tests for one-way and two-way ANOVA tests, respectively. T-tests were used for any independent two-group comparisons (experiments 3 and 4); Mann-Whitney U tests were used to compare ordinal NDS data between two groups (experiments 3 and 4). A significance level of $\alpha = 0.05$ was used to determine any significant results.

## Results

### Exclusions

There was no unexpected mortality in this study. Due to the nature of oral dosing and the motor impairments animals experienced, some animals did not consume their entire dose within the hour it was given. The majority of animals (75%) ate their dose within 15 minutes. We compared primary endpoints both including and excluding animals that took longer than 15 minutes to eat their dose, and there was no difference in our conclusions. Therefore, we continued our analysis using an "intention to treat" approach, including all animals in analysis.

In experiment 1, one animal's water and food measurements were excluded on day 6 due to a leaky water bottle. In experiment 2, two animals in the COLC-POST group and one animal in the COLC-PRE group were excluded from food measurements because they were given supplemental wet food to help recovery. One animal in the COLC-POST group was excluded from cell counts due to histological artefact in the region of interest. In experiment 4, two animals in the VEH group and four animals in the COLC-POST group were excluded from the ladder task on day 3 due to failure to cross the ladder apparatus.

### Experiment 1

In all animals, body weight continuously increased as expected due to the animals' young age. After 14 days of colchicine treatment in naïve rats, animals had gained an average of 13.2% of their baseline body weight (Fig 1A). Food and water consumption were consistent over the course of 14 days when compared to baseline consumption (Fig 1B and 1C). Feeding ranged from 79.3–125% of baseline, and drinking ranged from 79.5–128.2% of baseline.

The average RGS score at baseline was 0.4 (out of a possible maximum score of 8) compared to 0.6 on day 1 and 0.7 on day 14. Although these scores increased slightly, they are all well under the established analgesic intervention threshold of 2.68 (Fig 1D) [41].

Both temperature and activity remained consistent throughout the 14 days of colchicine dosing (Fig 1E and 1F). Temperature never exceeded ±1°C from baseline in any animal. Activity did not increase or decrease as dosing occurred.

### Experiment 2

There were no significant differences in NDS score at 48 hrs post-ICH (Fig 2A, p = 0.350). There were no significant differences between groups in food consumption (Fig 2B, p = 0.897) or water consumption (Fig 2C, p = 0.965). Additionally, muscle water content did not differ between groups (Fig 2D, p = 0.543), further supporting a lack of effect of colchicine on hydration. There was a significant effect of time on weight, with animals experiencing weight loss

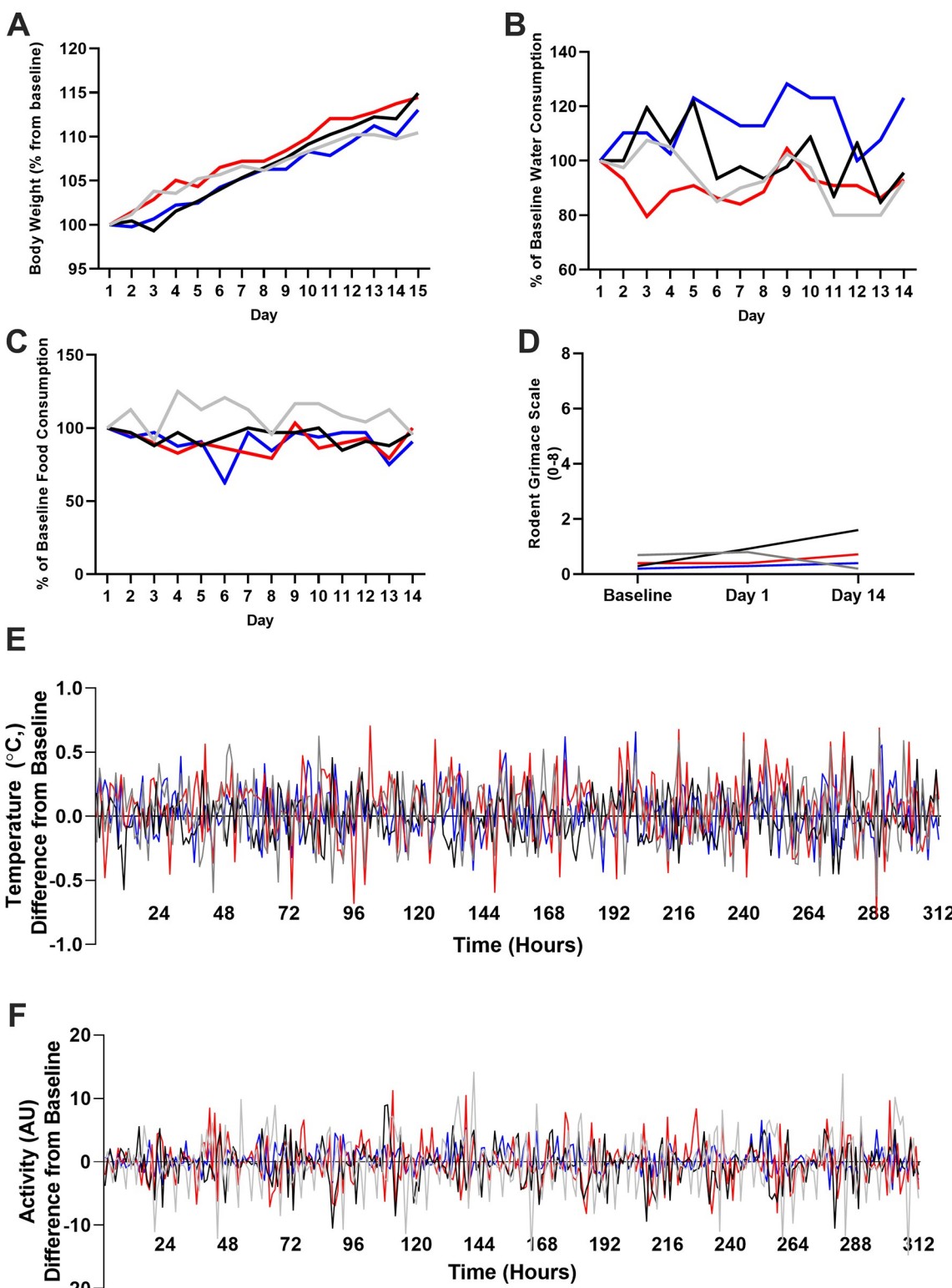

**Fig 1. Experiment 1 safety measures.** Body weight steadily increased in all animals (A) across experiment 1. Water (B) and food consumption (C) did not decrease by more than 20%, indicating little cause for concern. Pain (D) did not increase by more than 2 points in any animals, indicating the treatment was well tolerated. Body temperature (E) did not differ from baseline by more than 1˚C in any animals, at any point in time. Dosing did not affect observed activity levels (F). Temperature and activity are graphed as difference from baseline (averaged hourly to account for circadian rhythms), where a positive reading means that animals were hotter/more active post-treatment. Lines represent individual animals.

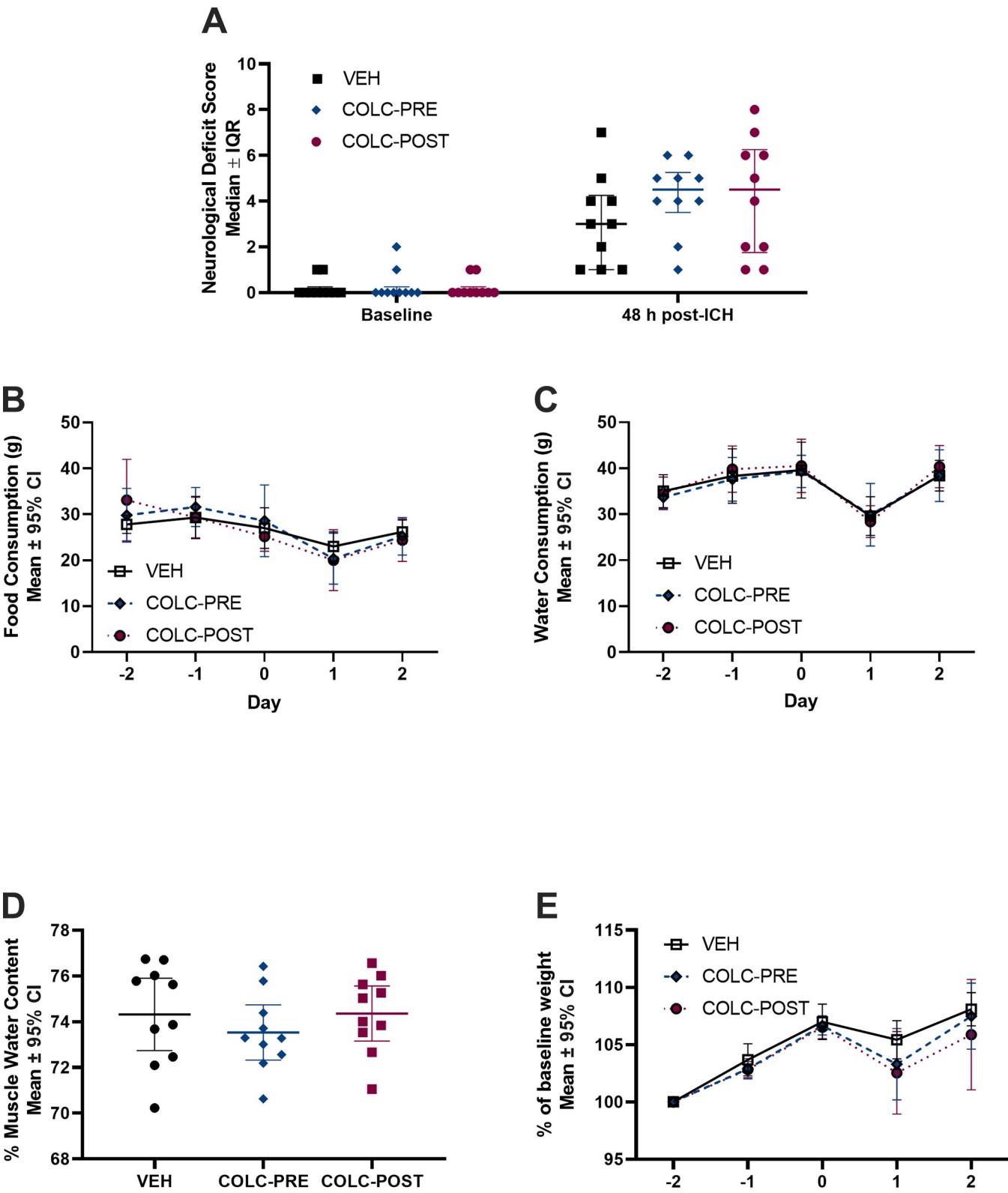

**Fig 2. Experiment 2.** ICH significantly worsened NDS at 48 hours post-ICH (A, p<0.001), but no differences were found among the three groups (p = 0.350). Food consumption (B, p = 0.897), water consumption (C, p = 0.965), and muscle water content (D, p = 0.543) did not differ between groups at any time assessed. ICH resulted in significant weight loss post-ICH (E, p<0.001), but no difference among groups were found at any time (p>0.999). n = 10/group.

after the ICH (Fig 2E, p<0.001). However, there were no differences in weight loss between the groups (p>0.999).

The collagenase model resulted in an average ICH volume of 40.7 mm$^3$. There was no difference in hematoma volume between groups (Fig 3A, p = 0.168), which was our primary endpoint in this experiment. We did not see a difference in cell counts in the PHZ between groups (Fig 3B, p = 0.071), indicating statistically equivalent amounts of cell death.

## Experiment 3

Although ICH caused neurological deficits (Fig 4A, p<0.001), there was no significant difference between the colchicine and vehicle groups (Fig 4A, p = 0.169).

There was a significant effect of region on brain water content. Higher brain water content, from serum extrusion and edema, was present in the ICH-damaged striatum (Fig 4B, region main effect p<0.001, ipsilateral striatum p<0.001 compared to contralateral striatum), and the overlying cortex (p<0.001 compared to contralateral cortex), but not in other regions such as the cerebellum, which serves as a control. There was no effect of colchicine treatment on any brain region (treatment main effect p = 0.412), which was our primary endpoint.

## Experiment 4

Neurological deficit score increased after ICH compared to baseline (Fig 5A, p<0.001). However, there was no difference in neurological deficits between groups at any time (all p>0.507). Walking ability, as assessed by the slip rate when crossing a horizontal ladder, decreased after ICH (Fig 5B, Time Effect, p = 0.016). There were no group differences in scores on the ladder task at any time (p = 0.791). When comparing composite rank scores taking into account NDS and ladder on all testing days, our primary endpoint, there were no significant differences in the average ranks between the colchicine and vehicle group (p = 0.439). Thus, colchicine was not beneficial. The ICH damage was contained within the striatum, and was typically located in the ventral and lateral region of the striatum. Total tissue lost at 14 days post-ICH did not differ between groups (Fig 5C, p = 0.985); thus, the treatment was not neuroprotective.

There was no significant difference in density of Perls positive cells between the groups (Fig 6A, p = 0.115), but the colchicine group did have a significantly reduced total volume of Perls positive cells (Fig 6B, p = 0.018) indicating a diminished inflammatory response.

## Discussion

In this study, low-dose colchicine did not adversely affect any indicators of general health (i.e., food and water consumption, body weight, temperature, general activity, and pain). This treatment also did not have any impact on bleeding volume, suggesting that colchicine will not worsen bleeding when given to ICH patients. Our study provides proof of concept that colchicine can reduce the perihematomal brain inflammatory response, however colchicine did not affect brain swelling, neurological impairments or brain injury in our model. This suggests that low-dose colchicine will not harm (e.g., worsen bleeding) patients with ICH, should their hemorrhage occur while already being treated with the drug or given the drug soon after the bleed. We found no suggestion that the reduced perihematomal inflammation observed with colchicine treatment translated to a meaningful improvement in neurological outcome (e.g., reduction in brain injury), however, this warrants further exploration in clinical trials of ICH patients receiving colchicine for the prevention of ischemic vascular events.

As with many negative findings, one must acknowledge that other drug dosing regimens might prove to be more effective (or harmful). Here, we used 0.05 mg/kg of colchicine once daily, administered orally via cookie dough, based on the commonly-used clinical dose of 5

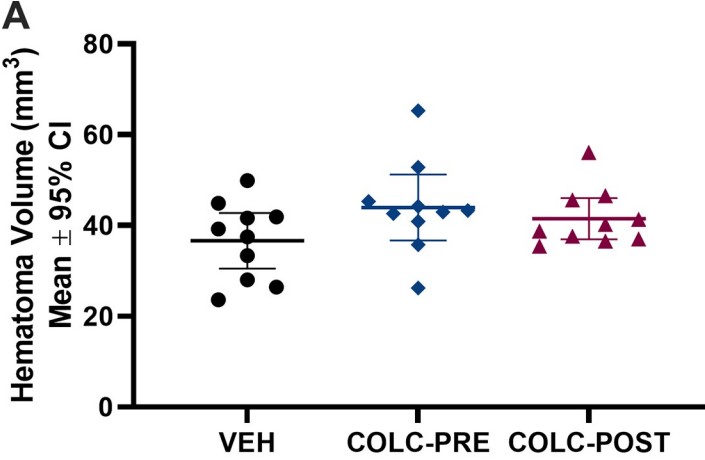

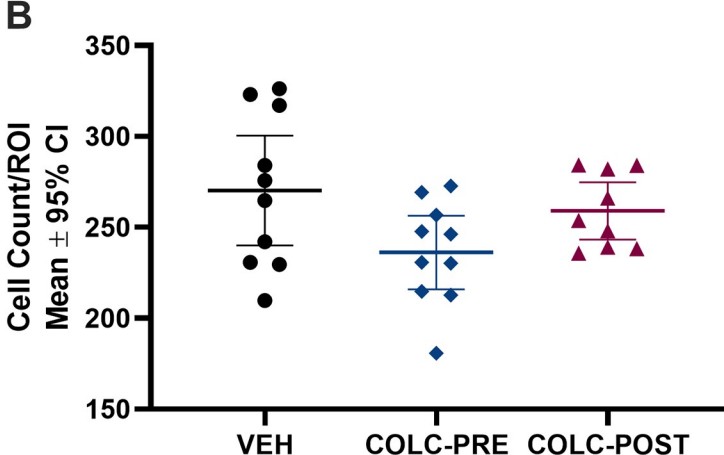

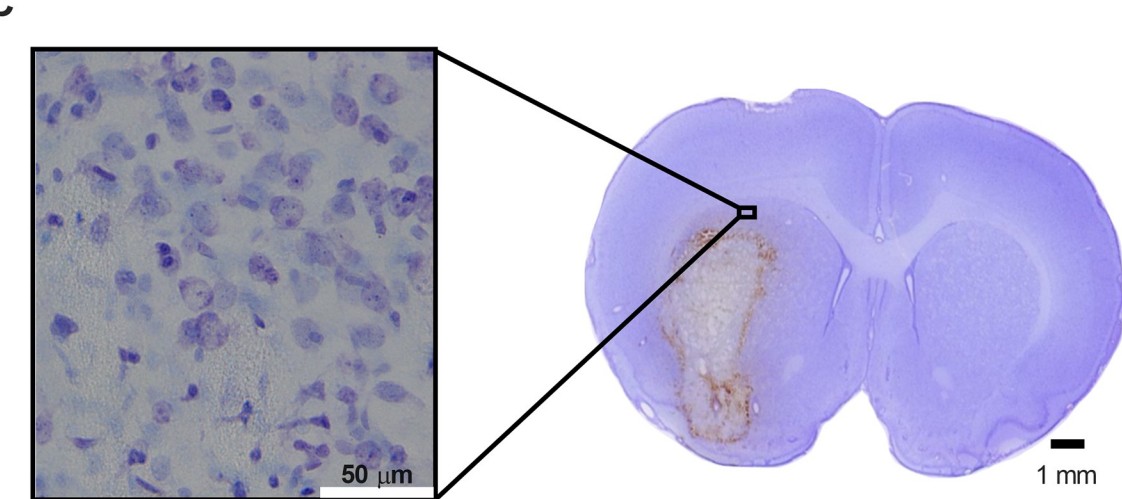

**Fig 3. Experiment 2 hematoma volume and cell counts.** No differences in hematoma volume (A, p = 0.168), or cell density in the peri-hematoma zone (B, p = 0.071) were found at 48 hours post-ICH. C. Representative image of neurons in the PHZ stained using cresyl violet. White scale bar represents 50 μm, black scale bar represents 1 mm.

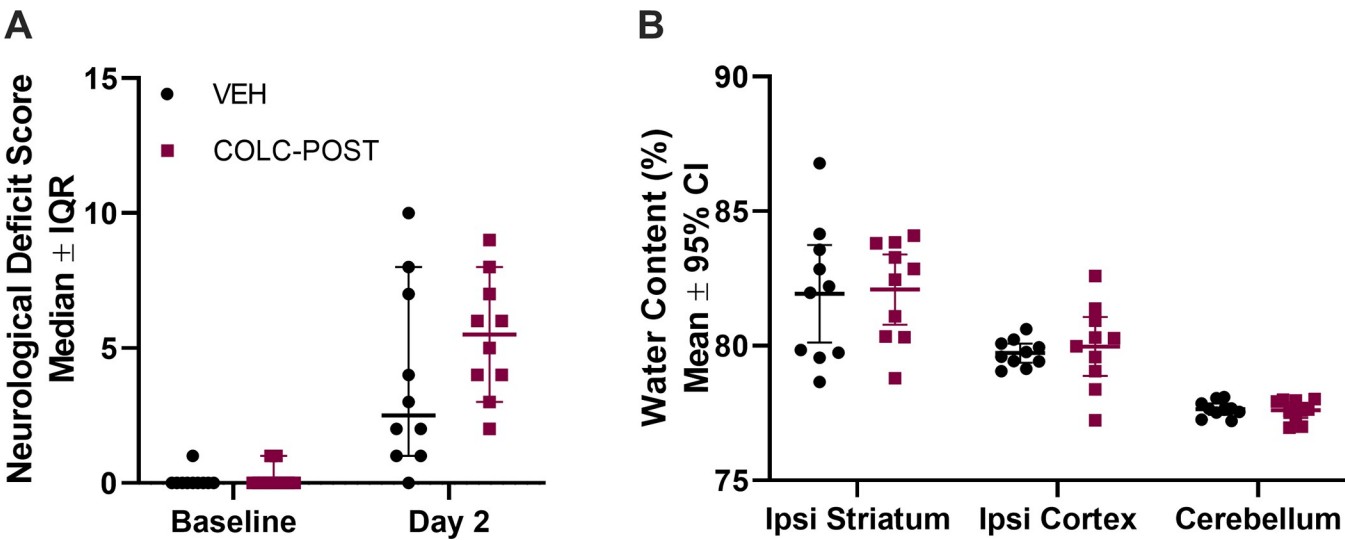

**Fig 4. Experiment 3 edema and behaviour.** ICH significantly worsened NDS scores at 48 hours post-ICH (A, p<0.001), but no differences between groups were found (p = 0.169). Brain water content at 48 hours post-stroke (B) was significantly increased in the ipsilateral striatum (p<0.001), but did not differ between groups (p = 0.412). n = 10/group.

mg [10, 27]. This dose of colchicine proved safe as it did not have any impact on body weight, food consumption, water consumption, hydration, body temperature, activity, and did not cause pain in our rats. We evaluated these because common side effects of colchicine in patients include abdominal pain, nausea, digestive issues (e.g., diarrhea, vomiting), and drug toxicity, at least at higher doses [42]. Therefore, our dose of colchicine seemed to be safe, but future studies considering higher or more frequent doses should re-evaluate these measures.

The inhibitory effects of colchicine on microtubule polymerization are largely responsible for its anti-inflammatory effects [43, 44]. Inhibiting microtubule polymerization further facilitates the inhibition of inflammatory cell proliferation, cell migration, and the nucleotide-binding domain (NOD)-like receptor protein 3 (NLRP3) inflammasome which has downstream effects on cytokine and chemokine production [44]. The broad range of colchicine's anti-inflammatory effects act on all stages of atherosclerotic plaque development, which make it a promising candidate for the prevention of cardiovascular disease and ischemic stroke [11, 44, 45].

The Low-Dose Colchicine 2 (LoDoCo2) trial included over 5500 patients with chronic coronary disease randomized to receive 0.5 mg/day of colchicine (P.O.) or placebo. That study found the relative risk of cardiovascular death, spontaneous myocardial infarction, ischemic stroke, or ischemia-driven coronary revascularization to be reduced by 31% in patients given the colchicine treatment [12]. Another large study, the COLCOT trial, randomized over 4500 patients to receive the same treatment regimen or placebo within 30 days of myocardial infarction. Colchicine treatment reduced the incidence of stroke, myocardial infarction, and cardiac arrest by approximately 48% [18]. A recent meta-analysis of 6 randomized controlled trials found that daily colchicine treatment significantly reduced the rate of ischemic stroke events (OR = 0.49) [10]. As such, colchicine is promising as a novel treatment option for the prevention of thromboembolic events in ICH patents.

It is important to note that colchicine has also been shown to inhibit platelet aggregation *in-vitro* when using blood from healthy subjects [17, 46–48]. This effect is almost exclusively demonstrated when platelets are stimulated by soluble-agonists (e.g. adenosine diphosphate,

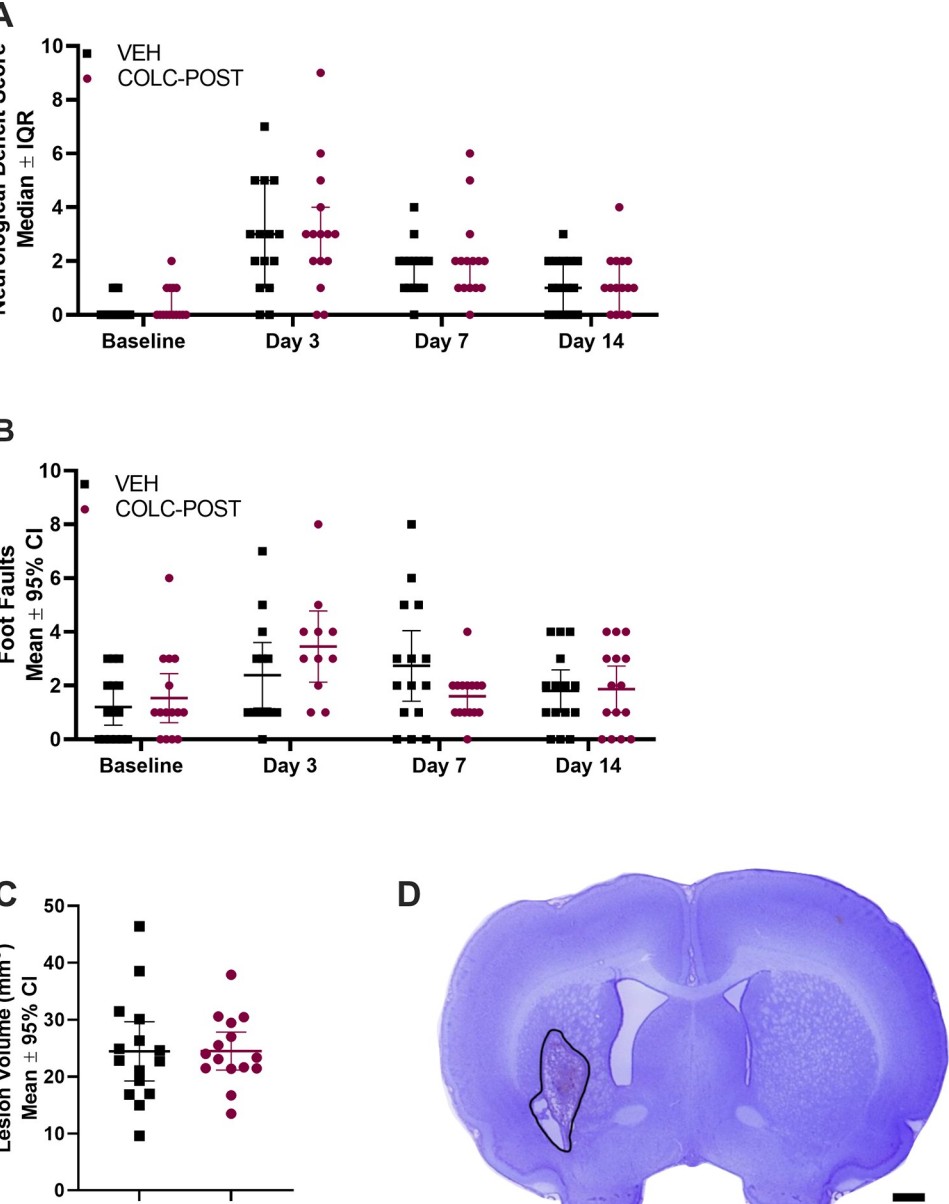

**Fig 5. Experiment 4 behaviour and lesion volume.** ICH significantly worsened NDS at all times post-ICH (A, p<0.001), and performance on the ladder test at day 3 only (B, p = 0.016). No differences in NDS (p>0.507) or ladder task (p = 0.791) were found between groups at any time. Lesion volume at 14 days post-stroke (C) was not affected by colchicine administration (p = 0.9848). D. Representative image of an animal from the colchicine group with a lesion volume of 21.47 mm$^3$. This animal demonstrates ventriculomegaly and the hematoma resolution at 14 days post-ICH. Scale bar represents 1 mm.

collagen, thrombin) [48]. While the mechanisms of colchicine's effect on platelet aggregation are unclear, a recent comprehensive review by Reddel et. al highlights that concentrations of colchicine much greater than those seen clinically can change platelet microtubular structure, which may influence platelet aggregation [48]. However, it is unknown if this structural change is clinically relevant, as it has not been investigated in patients or at clinically relevant doses/ concentrations. The inhibitory effect on platelet aggregation has mostly been shown in studies

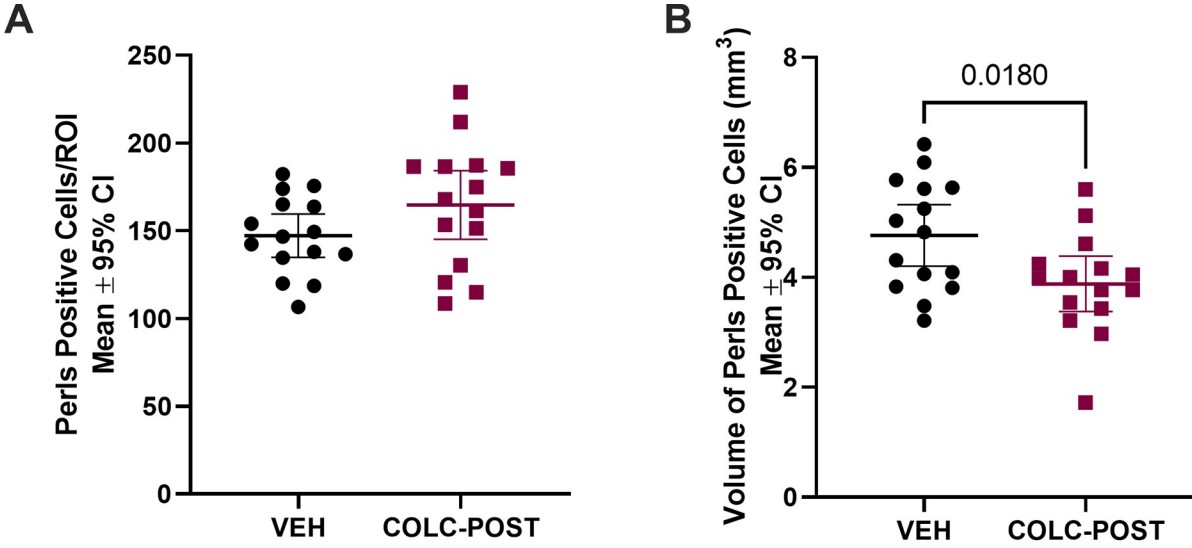

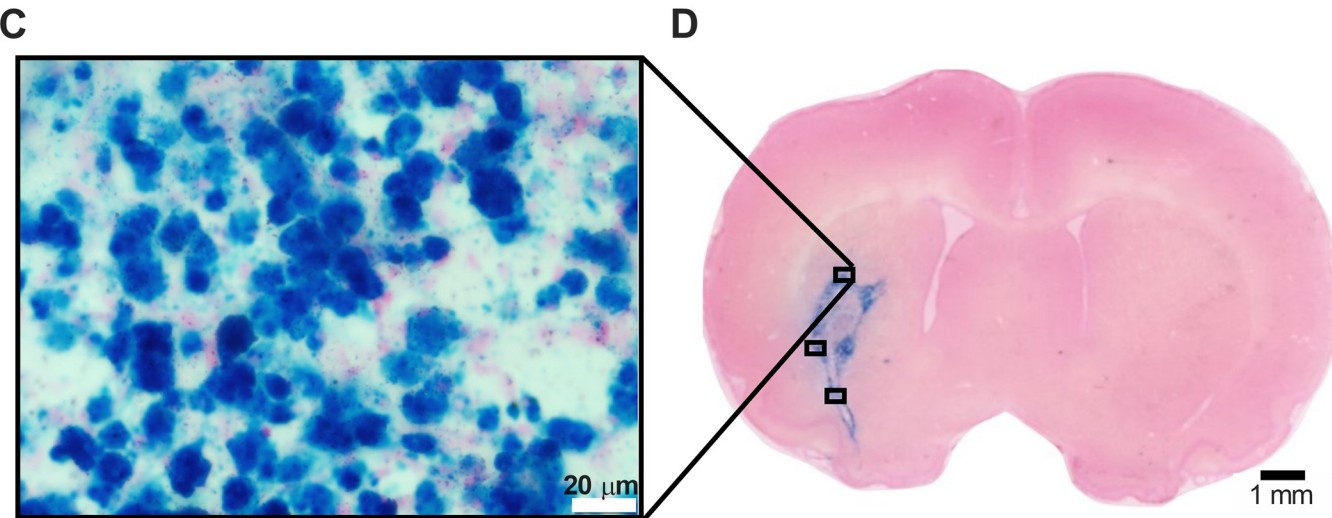

**Fig 6. Experiment 4 Perls staining.** Post-ICH colchicine administration did not affect the number of Perls positive cells per region of interest (A, p = 0.115), but did reduce the average volume of brain containing Perls positive cells (B, p = 0.018, average area of blue tissue multiplied by section interval and number of sections) at 14 days post-ICH. C. Perls positive cells under 40 × magnification. D. Representative image of Perls staining. White scale bar represents 20 μm, black scale bar represents 1 mm.

using concentrations of colchicine much higher than physiological plasma concentrations, and only for short durations (< 1 hour) [48]. The dose used in our study is designed to be safe and clinically relevant, meaning that the effects on platelet aggregation and hemostasis are likely minimal. Fittingly, there was no increased risk of bleeding outcomes observed with colchicine treatment in the COLCOT and LoDoCo2 randomized trials [12, 49]. However, it was still important to assess hematoma volume to ensure that this dose does not increase bleeding after ICH.

The intra-striatal collagenase infusion used in our study causes bleeding over several hours [19, 20], and is therefore a suitable model to evaluate bleeding risk, both with pre- and post-stroke therapies (e.g., coagulopathy [50]). As we did not find any differences in hematoma volume 48 hours after ICH among the COLC-PRE, COLC-POST, and VEH groups, it appears that any effects of colchicine on hemostasis are not significant enough to affect ICH volume. The COLC-PRE group began treatment two days before the ICH so as to model those under active treatment (cardiovascular disease, treatment of gout, pericarditis, etc. [43]) who then experience an ICH. In the COLC-POST group, treatment was initiated 2 hours after collagenase infusion so as to specifically test for any impact on bleeding on injury in this acute period. Our findings show that a low dose of colchicine is unlikely to increase bleeding in ICH patients that are taking colchicine prior to stroke or in those that are given colchicine treatment for secondary stroke prevention shortly after their bleed.

After ICH, we found no differences in brain water content between the colchicine and vehicle groups (experiment 3). This was important to assess because inflammation disrupts the blood brain barrier and allows for the infiltration of immune cells into the brain, which both exacerbate edema after ICH [14, 51]. Perihematomal edema, which begins within hours of ICH and peaks at ~10 days following ictus in humans, is thought to be caused by a series of neuroinflammatory responses associated with the formation and mechanical destruction of ICH, such as hemoglobin breakdown [51, 52]. Edema has been suggested to be a useful surrogate marker of secondary brain injury and an independent prognostic factor in ICH patients. Increasing perihematomal edema in the first days after an ICH has been associated with worsened outcomes. The lack of effect of colchicine on edema may be due to relatively weak anti-inflammatory actions of our low-dose treatment, and it is possible that multi-modal therapies, including colchicine, may be needed to effectively reduce edema. A final estimate of treatment efficacy will only be determined with additional data and meta-analyses. While colchicine did not affect edema, the drug demonstrated no negative effects on brain water content or neuronal cell loss, further suggesting this treatment is safe for acute post-ICH treatment.

In experiment 4, we found a significant decrease in the volume of iron-positive macrophages around the lesion in the COLC-POST group compared to the VEH group at 14 days after ICH, which suggests that inflammation was reduced in animals that were given colchicine. Despite this, colchicine did not provide any neuroprotective benefits after ICH. A limitation of our study is that we only used a single endpoint to assess the anti-inflammatory effects of colchicine. Had colchicine reduced the injury in our experimental model, additional investigation into the potential anti-inflammatory effects would have been necessary. In other preclinical studies, some anti-inflammatory treatments have reduced secondary injury through targeting inflammatory mechanisms, such as microglia activation and cytokine signaling [14, 53, 54]. If colchicine provides benefit in future studies, the effects on microglial activation, NLRP3 inflammasome activation and expression, and cytokine and chemokine expression should also be assessed [44, 45]. While these endpoints may provide interesting insights on colchicine's anti-inflammatory action, the absence of neuroprotective benefit provided by our dose did not warrant further investigation on inflammatory effects in this model.

Additional limitations include that some animals did not consume their dose of colchicine within the first hour that it was given to them. This only occurred during the first few days after ICH. Across all experiments, 75% of animals ate all their cookie dough within 15 minutes each time it was given. These differences in timing did not seem to affect our data as almost all animals (94%) consumed each dose fully before their next dose was given and excluding these animals from our analysis did not influence our results. Thus, we proceeded with an intention-to-treat analysis.

In summary, using a rat model of ICH, 0.05 mg/kg of colchicine given daily starting 2 hours after collagenase infusion did not impact bleeding, edema, cell death, lesion volume, body weight, hydration, food consumption, water consumption, pain, or neurological deficits despite reducing perihematomal inflammation. Starting this treatment 2 days before ICH (to model the patient population already taking colchicine prior to stroke) also did not impact these endpoints. Although we did not find any additional neuroprotective benefits, colchicine was well-tolerated and did not worsen any of our endpoint measures. It appears that daily oral administration of low-dose colchicine, a promising treatment for stroke prevention [10, 11, 18], is safe to use in an ICH setting. These important safety findings suggest that this treatment is unlikely to negatively affect outcome when given clinically for the prevention of thrombo-embolic events after an acute ICH. This is relevant to patients with ICH, who are at high risk for thromboembolic events with current treatment strategies. This risk is aggravated by the imposed cessation of antithrombotic medications for significant periods after hemorrhage. Data from a Swedish Stroke Register (Riksstroke) suggests that amongst patients on antith-rombotic medications at the timing of their ICH, only 39.9% resume their antiplatelet treat-ment within 1 year after an ICH, while the median time to the resumption of anticoagulants was estimated to be 101 days after the ICH onset [55]. Even within the setting of the Restart or Stop Antithrombotics Randomized Trial, comparing antiplatelet resumption with discontinu-ation in antithrombotic-related ICH survivors, the median timing to randomization (and anti-platelet resumption) was 76 days (IQR: 29–146) from ICH onset, despite trial eligibility allowing it to occur as soon as 24 hours post-ICH [56]. These findings highlight the need for novel treatments that modify the high risk of ischemic major vascular events in ICH survivors, particularly early following ICH where patients are not receiving antithrombotic medications. Colchicine for the Prevention of Vascular Events After an Acute Intracerebral Hemorrhage (CoVasc-ICH; NCT05159219) study is an ongoing randomized controlled clinical trial testing the hypothesis that anti-inflammatory therapy with colchicine can reduce major adverse car-diovascular events following an acute ICH and attenuate ICH-related inflammatory brain injury.

## Supporting information

**S1 Data.**
(XLSX)

## Acknowledgments

We would like to thank Anna Kalisvaart and Sherry Gu for their help with this experiment.

## Author Contributions

**Conceptualization:** Aristeidis H. Katsanos, Ashkan Shoamanesh.

**Data curation:** Cassandra M. Wilkinson, Noam H. Sander.

**Formal analysis:** Cassandra M. Wilkinson, Tiffany F. C. Kung, Frederick Colbourne.

**Funding acquisition:** Aristeidis H. Katsanos, Ashkan Shoamanesh.

**Investigation:** Cassandra M. Wilkinson, Noam H. Sander.

**Methodology:** Cassandra M. Wilkinson, Tiffany F. C. Kung.

**Project administration:** Frederick Colbourne.

**Supervision:** Cassandra M. Wilkinson, Frederick Colbourne.

**Writing – original draft:** Cassandra M. Wilkinson, Noam H. Sander.

**Writing – review & editing:** Cassandra M. Wilkinson, Aristeidis H. Katsanos, Noam H. Sander, Tiffany F. C. Kung, Frederick Colbourne, Ashkan Shoamanesh.

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
