## [Decision Letter · Decision Letter 0]

30 Aug 2022

PONE-D-22-19690Colchicine pre-treatment and post-treatment does not worsen bleeding or functional outcome after collagenase-induced intracerebral hemorrhagePLOS ONE

Dear Dr. Colbourne,

Thank you for submitting your manuscript to PLOS ONE. After careful consideration, we feel that it has merit but does not fully meet PLOS ONE’s publication criteria as it currently stands. Therefore, we invite you to submit a revised version of the manuscript that addresses the points raised during the review process.

Specifically:Please elaborate on the rationale for not including a control group in Experiment 1.Please elaborate on the mechanism of the anti-inflammatory action of colchicine. Additional data using further parameters for neuroinflammation should be considered.As pointed out by both reviewers, the interactions of colchicine with platelet function should be elaborated more in detail. It would be useful to add information on the effect of colchicine on hemostasis and platelet function in your set-up.  Please also add discussion about the potential risks/benefits of colchicine in comparison to standard therapies to reduce post-ICH ischemic vascular events.Please provide a short summary and conclusion at the end of discussion.Please submit your revised manuscript by Oct 14 2022 11:59PM. If you will need more time than this to complete your revisions, please reply to this message or contact the journal office at plosone@plos.org. Please include the following items when submitting your revised manuscript:A rebuttal letter that responds to each point raised by the academic editor and reviewer(s). You should upload this letter as a separate file labeled 'Response to Reviewers'.A marked-up copy of your manuscript that highlights changes made to the original version. You should upload this as a separate file labeled 'Revised Manuscript with Track Changes'.An unmarked version of your revised paper without tracked changes. You should upload this as a separate file labeled 'Manuscript'.

We look forward to receiving your revised manuscript.

Kind regards,

Anna-Leena Sirén

Academic Editor

PLOS ONE

Journal Requirements:

Reviewers' comments:

Reviewer's Responses to Questions

**Comments to the Author**

1. Is the manuscript technically sound, and do the data support the conclusions?

Reviewer #1: Yes

Reviewer #2: Yes

2. Has the statistical analysis been performed appropriately and rigorously? 

Reviewer #1: Yes

Reviewer #2: Yes

3. Have the authors made all data underlying the findings in their manuscript fully available?

Reviewer #1: Yes

Reviewer #2: Yes

4. Is the manuscript presented in an intelligible fashion and written in standard English?

Reviewer #1: Yes

Reviewer #2: Yes

5. Review Comments to the Author

Reviewer #1: The authors assessed the safety and efficacy of colchicine in a rat model of acute collagenase-induced ICH and its potential role as a neuroprotective agent following experimental ICH. Most of the limitations have already been discussed adequately. However, the manuscript might be improved through the following constructive feedback.

1. Colchicine is a known modulator of platelet function. The effect and risks of colchicine as a platelet function modulator in intracerebral (striatal) hemorrhage could be more focused and discussed.

2. Minimizing the risk of ischemic events with anithrombotic therapy and otherwise risking a re-bleedin/progress of bleeding using antithrombotic therapy is usually hard to balance regarding the patient's treatment path. Is there any benefit of using colchicine instead of low-dose heparine or an early reonset of aspirin/clopidogrel therapy (e.g. 4 weeks after intracerebral hemorrhage) in reducing the risk of ischemic vascular events?

3. It would be an asset to have a conclusion at the end of the mauscript

Reviewer #2: This is an experimental study evaluating the effect of colchicine in a rat model of ICH. The authors evaluated safety and feasibility in their model of kollagenase induced ICH. The manuscript is well written and correctly reports findings according to the ARRIVE guidelines.

The main finding is that colchicine reduced the number of iron containing macophages in the perihemorrhagic zone; the authors then conclude that Colchicine treatment reduces inflammatory changes post ICH and may therefore constitue a therapeutic option after ICH.

Comments:

The abbreviation NDS is not introduced at first mention.

P14/Line 302: "The average score at baseline........" - please specify which score (grimace score?)

Experiment 1: Was there a control group? If yes, please report the findings. If not, what was the rationale?

The authors mention that colchicine may influence platelet function; is there any data about hemostasis/ platelet function in the current study? Bleeding time? Other platelet function tests?

The authors state that colchicine may have different anti-inflammatory effects, e. g. via influencing chemokine, cytokine expression, inflammatory cell proliferation and migration and has been shown to affect the NLRP3 inflammasome. In the present study they assessed PErls positive cells, i.e. cells containing iron residues as only marker for neuroinflammation. Given that drug treatment reduced this parameter I would suggest to further elaborate on the possible anti-inflammatory effect, e. g. by assessing microglia activation and other components of the inflammasome to further elaborate on this finding. This would substantially strenghten their conclusions.

6. PLOS authors have the option to publish the peer review history of their article (what does this mean?). If published, this will include your full peer review and any attached files.

Reviewer #1: No

Reviewer #2: No

---

## [Author Response · Author response to Decision Letter 0]

14 Sep 2022

We wish to thank the reviewers for their constructive comments. We have addressed their concerns as detailed below. We believe this has significantly improved our manuscript.

Reviewer #1: The authors assessed the safety and efficacy of colchicine in a rat model of acute collagenase-induced ICH and its potential role as a neuroprotective agent following experimental ICH. Most of the limitations have already been discussed adequately. However, the manuscript might be improved through the following constructive feedback.

1. Colchicine is a known modulator of platelet function. The effect and risks of colchicine as a platelet function modulator in intracerebral (striatal) hemorrhage could be more focused and discussed.

Response: We appreciate your suggestion to discuss colchicine as a platelet function modulator. To date, no studies have investigated the effects of colchicine on platelet function in ICH. However, the effects of colchicine on platelet function are well known (as you mentioned) and have been considerably investigated in-vitro. Thus, we have added further discussion (page 19, lines 470-484). We have also noted that colchicine is a known modulator of platelet function to improve the accuracy of the sentence in our introduction (page 4, lines 56-57)

2. Minimizing the risk of ischemic events with anithrombotic therapy and otherwise risking a re-bleedin/progress of bleeding using antithrombotic therapy is usually hard to balance regarding the patient's treatment path. Is there any benefit of using colchicine instead of low-dose heparine or an early reonset of aspirin/clopidogrel therapy (e.g. 4 weeks after intracerebral hemorrhage) in reducing the risk of ischemic vascular events?

Response: For some time after an ICH, anti-thrombotics (anticoagulants and antiplatelets) are usually not administered due to re-bleeding risks. Although the net benefit of colchicine has yet to be established in this setting, the Colchicine for the Prevention of Vascular Events After an Acute Intracerebral Hemorrhage (CoVasc-ICH; NCT05159219) study is an ongoing randomized controlled clinical trial from our group testing the hypothesis that anti-inflammatory therapy with colchicine can reduce major adverse cardiovascular events following an acute ICH and attenuate ICH-related inflammatory brain injury. This information, with corresponding references were added in our revised manuscript (page 22, lines 578-590).

3. It would be an asset to have a conclusion at the end of the mauscript

Response: We have now revised our final paragraph to include a summary of the present findings, our conclusions, and a final explanation of the implications/relevance of this study. (pages 21- 22, lines 545-594).

Reviewer #2: This is an experimental study evaluating the effect of colchicine in a rat model of ICH. The authors evaluated safety and feasibility in their model of kollagenase induced ICH. The manuscript is well written and correctly reports findings according to the ARRIVE guidelines.

The main finding is that colchicine reduced the number of iron containing macophages in the perihemorrhagic zone; the authors then conclude that Colchicine treatment reduces inflammatory changes post ICH and may therefore constitute a therapeutic option after ICH.

Comments:

The abbreviation NDS is not introduced at first mention. 

Response: The abbreviation NDS is now introduced at first mention (page 7, line 141): “Neurological deficit scale (NDS) and horizontal walking test scores were combined into a composite behavioural score as our primary endpoint for this experiment.”

P14/Line 302: "The average score at baseline........" - please specify which score (grimace score?)

Response: We have modified this sentence to specify the score. This sentence now reads “The average RGS score at baseline was 0.4 (out of a possible maximum score of 8) compared to 0.6 on day 1 and 0.7 on day 14.” (page 14, line 311)

Experiment 1: Was there a control group? If yes, please report the findings. If not, what was the rationale? 

Response: In an effort to reduce animal numbers, we did not have a separate control group for experiment 1. Including such a control group iwas unnecessary because baseline measurements were obtained before treatment was initiated. As well, the endpoints of this experiment are all very well established (e.g., expected weight gain and normal temperature ranges). This justification has been added to the manuscript (page 6, lines 120-121: “This experiment did not include a control group as the expected parameters are well known in healthy animals, and baseline measures were obtained for reference.”).

The authors mention that colchicine may influence platelet function; is there any data about hemostasis/ platelet function in the current study? Bleeding time? Other platelet function tests? 

Response: The effects of colchicine on blood clotting is a concern, but studies that have found significant effects on hemostasis and platelet function have mostly used doses of colchicine much higher than those that are safe to achieve clinically. Because our study used a low-dose of colchicine, specifically chosen to be safe and clinically-relevant, it is unlikely that we would have found a significant effect on hemostasis with our dosing regimen. Thus, we did not perform any tests of platelet function or assess peripheral bleeding time (e.g., tail bleed). However, any biologically-meaningful effects that colchicine on hemostasis should have been detected by group comparisons of intracerebral hematoma volume (most biologically relevant endpoint for this study), which were not significant. A similar explanation, along with additional discussion on the effects of colchicine on platelet function and aggregation, have now been added (discussion, page 19, lines 470-484).

The authors state that colchicine may have different anti-inflammatory effects, e. g. via influencing chemokine, cytokine expression, inflammatory cell proliferation and migration and has been shown to affect the NLRP3 inflammasome. In the present study they assessed PErls positive cells, i.e. cells containing iron residues as only marker for neuroinflammation. Given that drug treatment reduced this parameter I would suggest to further elaborate on the possible anti-inflammatory effect, e. g. by assessing microglia activation and other components of the inflammasome to further elaborate on this finding. This would substantially strenghten their conclusions. 

Response: Given that colchicine did not affect any of our primary endpoints of treatment efficacy (e.g., edema) and safety (e.g., hematoma volume), we can infer that any effects on the inflammatory response were inconsequential to stroke outcome. Notably, the impact of colchicine on inflammation was not sufficient to impact brain edema, cell death, or functional recovery. Had we found positive findings on any of those key, widely-accepted endpoints, we would have investigated the inflammatory response further to document the changes and to better identify important mechanisms of action. As suggested by the reviewer, we have added text to our discussion (page 21, lines 527-537) to acknowledge this, as well as to mention those additional endpoints (e.g., cytokine and chemokine expression) that could be assessed to gain further insight into the inflammatory effects of colchicine should, for instance, other colchicine treatment regimens provide benefit against ICH in animals or patients.

---

## [Decision Letter · Decision Letter 1]

6 Oct 2022

Colchicine pre-treatment and post-treatment does not worsen bleeding or functional outcome after collagenase-induced intracerebral hemorrhage

PONE-D-22-19690R1

Dear Dr. Colbourne,

We’re pleased to inform you that your manuscript has been judged scientifically suitable for publication and will be formally accepted for publication once it meets all outstanding technical requirements.

Kind regards,

Anna-Leena Sirén

Academic Editor

PLOS ONE

Additional Editor Comments (optional):

Reviewers' comments:

Reviewer's Responses to Questions

**Comments to the Author**

1. If the authors have adequately addressed your comments raised in a previous round of review and you feel that this manuscript is now acceptable for publication, you may indicate that here to bypass the “Comments to the Author” section, enter your conflict of interest statement in the “Confidential to Editor” section, and submit your "Accept" recommendation.

Reviewer #1: All comments have been addressed

2. Is the manuscript technically sound, and do the data support the conclusions?

Reviewer #1: Yes

3. Has the statistical analysis been performed appropriately and rigorously? 

Reviewer #1: Yes

4. Have the authors made all data underlying the findings in their manuscript fully available?

Reviewer #1: Yes

5. Is the manuscript presented in an intelligible fashion and written in standard English?

Reviewer #1: Yes

6. Review Comments to the Author

Reviewer #1: (No Response)

7. PLOS authors have the option to publish the peer review history of their article (what does this mean?). If published, this will include your full peer review and any attached files.

Reviewer #1: No

---

## [Editor Report · Acceptance letter]

7 Oct 2022

PONE-D-22-19690R1 

Colchicine pre-treatment and post-treatment does not worsen bleeding or functional outcome after collagenase-induced intracerebral hemorrhage 

Dear Dr. Colbourne:

I'm pleased to inform you that your manuscript has been deemed suitable for publication in PLOS ONE. Congratulations! Your manuscript is now with our production department. 

Kind regards, 

on behalf of

Dr. Anna-Leena Sirén 

Academic Editor

PLOS ONE